# Enhancing Scalability of Pre-trained Language Models via Efficient Parameter Sharing

**Peiyu Liu**[1,2][*], **Ze-Feng Gao**[1,3*], **Yushuo Chen**[1,2], **Wayne Xin Zhao**[1,2†] and **Ji-Rong Wen**[1,2]

[1]Gaoling School of Artificial Intelligence, Renmin University of China
[2]Beijing Key Laboratory of Big Data Management and Analysis Methods
[3]Department of Physics, Renmin University of China
{liupeiyustu,zfgao,chenyushuo,jrwen}@ruc.edu.cn,
batmanfly@gmail.com

## Abstract

In this paper, we propose a highly parameter-efficient approach to scaling pre-trained language models (PLMs) to a deeper model depth. Unlike prior work that shares all parameters or uses extra blocks, we design a more capable parameter-sharing architecture based on matrix product operator (MPO), an efficient tensor decomposition method to factorize the parameter matrix into a set of local tensors. Based on such a decomposition, we share the important local tensor across all layers for reducing the model size and meanwhile keep layer-specific tensors (also using Adapters) for enhancing the adaptation flexibility. To improve the model training, we further propose a stable initialization algorithm tailored for the MPO-based architecture. Extensive experiments have demonstrated the effectiveness of our proposed model in enhancing scalability and achieving higher performance (*i.e.,* with fewer parameters than BERT_{BASE}, we successfully scale the model depth by a factor of $4\times$ and even achieve 0.1 points higher than BERT_{LARGE} for GLUE score). The code to reproduce the results of this paper can be found at https://github.com/RUCAIBox/MPOBERT-code.

## 1 Introduction

Recently, pre-trained language models (PLMs) have achieved huge success in a variety of NLP tasks by exploring ever *larger* model architecture (Raffel et al., 2020; Radford et al., 2019). It has been shown that there potentially exists a scaling law between the model size and model capacity for PLMs (Kaplan et al., 2020), attracting many efforts to enhance the performance by scaling model size (Chowdhery et al., 2022; Wang et al., 2022b).

As a straightforward approach, we can directly increase the layer number of networks for improving the model capacity (Wang et al., 2022b; Huang

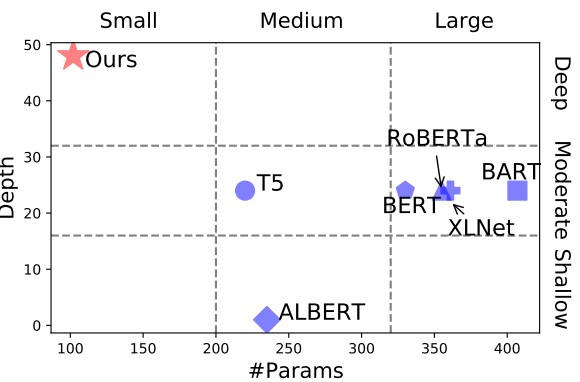

Figure 1: A comparison of our model and representative PLMs in the dimensions of *model size* and *model depth*.

et al., 2020). While, a very deep architecture typically corresponds to a significantly large model size, leading to high costs in both computation and storage (Gong et al., 2019). And, it is difficult to deploy deep networks in resource-limited settings, though it usually has a stronger model capacity. Therefore, there is an urgent need for developing a parameter-efficient way for scaling the model depth.

To reduce the parameters in deep networks, weight sharing has proven to be very useful to design lightweight architectures (Zhang et al., 2022; Lan et al., 2019). As a representative one by across-layer parameter sharing, ALBERT (Lan et al., 2019) keeps only ten percent of the whole parameters of BERT while maintaining comparable performance. Although the idea of parameter sharing is simple yet (to some extent) effective, it has been found that identical weights across different layers are the main cause of performance degradation (Zhang et al., 2022). To address this issue, extra blocks are designed to elevate parameter diversity in each layer (Nouriborji et al., 2022). While they still use the rigid architecture of shared layer weights, having a limited model capacity. Besides, it is difficult to optimize very deep models,

---

[*]Authors contributed equally.
[†]Corresponding author.

especially when shared components are involved. Although recent studies (Wang et al., 2022b; Huang et al., 2020) propose improved initialization methods, they do not consider the case with parameter sharing, thus likely leading to a suboptimal performance on a parameter-sharing architecture.

To address these challenges, in this paper, we propose a highly parameter-efficient approach to scaling PLMs to a deeper model architecture. As the core contribution, we propose a *matrix product operator* (MPO) based parameter-sharing architecture for deep Transformer networks. Via MPO decomposition, a parameter matrix can be decomposed into *central tensors* (containing the major information) and *auxiliary tensors* (containing the supplementary information). Our approach shares the central tensors of the parameter matrices across all layers for reducing the model size, and meanwhile keeps layer-specific auxiliary tensors for enhancing the adaptation flexibility. In order to train such a deep architecture, we propose an MPO-based initialization method by utilizing the MPO decomposition results of ALBERT. Further, for the auxiliary tensors of higher layers (more than 24 layers in ALBERT), we propose to set the parameters with scaling coefficients derived from theoretical analysis. We theoretically show it can address the training instability regardless of the model depth.

Our work provides a novel parameter-sharing way for scaling model depth, which can be generally applied to various Transformer-based models (Zhao et al., 2023). We conduct extensive experiments to evaluate the performance of the proposed model on the GLUE benchmark in comparison to PLMs with varied model sizes (tiny, small and large). Experiments results have demonstrated the effectiveness of the proposed model in reducing the model size and achieving competitive performance. With fewer parameters than BERT$_{BASE}$, we scale the model depth by a factor of 4x and achieve 0.1 points higher than BERT$_{LARGE}$ for GLUE score.

## 2 Related Work

**Matrix Product Operators**. Matrix product operators (*a.k.a.* tensor-train operators (Oseledets, 2011)) were proposed for a more effective representation of the linear structure of neural networks (Gao et al., 2020a), which was used to compress deep neural networks (Novikov et al., 2015), convolutional neural networks (Garipov et al., 2016; Yu et al., 2017), and LSTM (Gao et al.,

2020b; Sun et al., 2020a). Based on MPO decomposition, recent studies designed lightweight fine-tuning and compression methods for PLMs (Liu et al., 2021), developed parameter-efficient MoE architecture (Gao et al., 2022), over-parametrization PLMs (Gao et al., 2023) and empirical study the emergency ability in quantized large language models (Liu et al., 2023). Unlike these works, our work aims to develop a very deep PLM with lightweight architecture and stable training.

**Parameter-Efficient PLMs**. Existing efforts to reduce the parameters of PLMs can be broadly categorized into three major lines: knowledge distillation, model pruning, and parameter sharing. For knowledge distillation-based methods (Sanh et al., 2019; Sun et al., 2020b,b; Liu et al., 2020; Wang et al., 2022a), PLMs were distilled into student networks with much fewer parameters. For pruning-based methods, they tried to remove less important components (Michel et al., 2019; Wang et al., 2020) or very small weights (Chen et al., 2020). Moreover, the parameter-sharing method was further proposed by sharing all parameters (Lan et al., 2019) or incorporating specific auxiliary components (Reid et al., 2021; Nouriborji et al., 2022). Different from these works, we design an MPO-based architecture that can reduce the model size and enable adaptation flexibility, by decomposing the original matrix.

**Optimization for Deep Models**. Although it is simple to increase the number of layers for scaling model size, it is difficult to optimize very deep networks due to the training instability issue. Several studies have proposed different strategies to overcome this difficulty for training deep Transformer networks, including Fixup (Zhang et al., 2019) by properly rescaling standard initialization, T-Fixup (Huang et al., 2020) by proposing a weight initialization scheme, and DeepNorm (Wang et al., 2022b) by introducing new normalization function. As a comparison, we study how to optimize the deep MPO-based architecture with the parameter sharing strategy, and explore the use of well-trained PLMs for initialization, which has a different focus from existing work.

## 3 Method

In this section, we describe the proposed *MPOBERT* approach for building deep PLMs via a highly parameter-efficient architecture. Our ap-

proach follows the classic *weight sharing* paradigm while introducing a principled mechanism for sharing informative parameters across layers and also enabling layer-specific weight adaptation.

## 3.1 Overview of Our Approach

Although weight sharing has been widely explored for building compact PLMs (Lan et al., 2019), existing studies either share all the parameters across layers (Lan et al., 2019) or incorporate additional blocks to facilitate the sharing (Zhang et al., 2022; Nouriborji et al., 2022). They either have limited model capacity with a rigid architecture or require additional efforts for maintenance.

Considering the above issues, we motivate our approach in two aspects. Firstly, only informative parameters should be shared across layers, instead of all the parameters. Second, it should not affect the capacity to capture layer-specific variations. To achieve this, we utilize the MPO decomposition (Liu et al., 2021) to develop a parameter-efficient architecture by sharing informative components across layers and keeping layer-specific supplementary components (Section 3.2). As another potential issue, it is difficult to optimize deep PLMs due to unstable training (Wang et al., 2022b), especially when weight sharing (Lan et al., 2019) is involved. We further propose a simple yet effective method to stabilize the training of PLMs (Section 3.3). Next, we introduce the technical details of our approach.

## 3.2 MPO-based Transformer Layer

In this section, we first introduce the MPO decomposition and introduce how to utilize it for building parameter-efficient deep PLMs.

### 3.2.1 MPO Decomposition

Given a weight matrix $\mathbf{W} \in \mathbb{R}^{I \times J}$, MPO decomposition can decompose a matrix into a product of $n$ tensors by reshaping the two dimension sizes $I$ and $J$:

$$\mathbf{W}_{i_1,...,i_n,j_1,...,j_n} = \mathcal{T}^{(1)}[i_1, j_1] \cdots \mathcal{T}^{(n)}[i_n, j_n], \quad (1)$$

where we have $I = \prod_{k=1}^{n} i_k$, $J = \prod_{k=1}^{n} j_k$, and $\mathcal{T}^{(k)}[i_k, j_k]$ is a 4-dimensional tensor with size $d_{k-1} \times i_k \times j_k \times d_k$ in which $d_k$ is a bond dimension linking $T^{(k)}$ and $T^{(k+1)}$ with $d_0 = d_n = 1$. For simplicity, we omit the bond dimensions in Eq. (4). When $n$ is odd, the middle tensor contains the most parameters (with the largest bond dimensions), while the parameter sizes of the rest

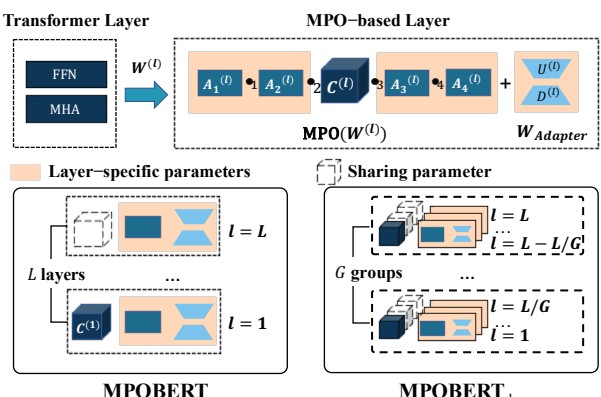

Figure 2: Overview architecture of MPOBERT and MPOBERT$_+$. We use blocks with dashed borderlines to represent shared central tensors. Central tensors are shared across all $L$ Layers in MPOBERT and within groups in MPOBERT$_+$.

decrease with the increasing distance to the middle tensor. Following Gao et al. (2022), we further simplify the decomposition results of a matrix as a central tensor $\mathcal{C}$ (the middle tensor) and auxiliary tensors $\{\mathcal{A}_i\}_{i=1}^{n-1}$ (the rest tensor).

As a major merit, such a decomposition can effectively reorganize and aggregate the information of the matrix (Liu et al., 2021): central tensor $\mathcal{C}$ can encode the essential information of the original matrix, while auxiliary tensors $\{\mathcal{A}_i\}_{i=1}^{n-1}$ serve as its complement to exactly reconstruct the matrix.

### 3.2.2 MPO-based Scaling to Deep Models

Based on MPO decomposition, the essence of our scaling method is to share the central tensor across layers (*capturing the essential information*) and keep layer-specific auxiliary tensors (*modeling layer-specific variations*). Fig. 2 shows the overview architecture of the proposed MPOBERT.

**Cross-layer Parameter Sharing**. To introduce our architecture, we consider a simplified structure of $L$ layers, each consisting of a single matrix. With the five-order MPO decomposition (*i.e.*, $n = 5$), we can obtain the decomposition results for a weight matrix ($\mathbf{W}^{(l)}$), denoted as $\{\mathcal{C}^{(l)}, \mathcal{A}_1^{(l)}, \mathcal{A}_2^{(l)}, \mathcal{A}_3^{(l)}, \mathcal{A}_4^{(l)}\}_{l=1}^{L}$, where $\mathcal{C}^{(l)}$ and $\{\mathcal{A}_i^{(l)}\}_{i=1}^{4}$ are the central tensor and auxiliary tensors of the $l$-th layer. Our approach is to set a shared central tensor $\mathcal{C}$ across layers, which means that $\mathcal{C}^{(l)} = \mathcal{C}$ ($\forall l = 1 \cdots L$). As shown in Appendix A.1, the central tensor contains the major proportion of parameters (more than 90%), and thus our method can largely reduce the parame-

ters when scaling a PLM to very deep architecture. We implement our proposed efficient parameter-sharing strategy upon the BERT (Devlin et al., 2018), named *MPOBERT*, which shares the central tensor across layers. Note that this strategy can be easily applied to multiple matrices in a Transformer layer, and we omit the discussion for the multi-matrix extension. Another extension is to share the central tensor by different grouping layers. We implement a layer-grouping strategy upon the BERT (Devlin et al., 2018), named *MPOBERT+*, which divides the layers into multiple parts and sets unique shared central tensors in each group.

**Layer-specific Weight Adaptation**. Unlike AL-BERT (Lan et al., 2019), our MPO-based architecture enables layer-specific adaptation by keeping layer-specific auxiliary tensors ($\{\mathcal{A}_i^{(l)}\}_{i=1}^4$). These auxiliary tensors are decomposed from the original matrix, instead of extra blocks (Zhang et al., 2022). They only contain a very small proportion of parameters, which does not significantly increase the model size. While, another merit of MPO decomposition is that these tensors are highly correlated via bond dimensions, and a small perturbation on an auxiliary tensor can reflect the whole matrix (Liu et al., 2021). If the downstream task requires more layer specificity, we can further incorporate low-rank adapters (Hu et al., 2021) for layer-specific adaptation. Specifically, we denote $\mathbf{W}_{Adapter}^{(l)}$ as the low-rank adapter for $\mathbf{W}^{(l)}$. In this way, $\mathbf{W}^{(l)}$ can be formulated as a set of tensors: $\{\mathcal{C}^{(l)}, \mathcal{A}_1^{(l)}, \mathcal{A}_2^{(l)}, \mathcal{A}_3^{(l)}, \mathcal{A}_4^{(l)}, \mathbf{W}_{Adapter}^{(l)}\}$. The parameter scale of adapters, $L \times r \times d_{total}$, is determined by the layer number $L$, the rank $r$, and the shape of the original matrix ($d_{total} = d_{in} + d_{out}$ is the sum of the input and output dimensions of a Transformer Layer). Since we employ low-rank adapters, we can effectively control the number of additional parameters from adapters.

## 3.3 Stable Training for MPOBERT

With the above MPO-based approach, we can scale a PLM to a deeper architecture in a highly parameter-efficient way. However, as shown in prior studies (Lan et al., 2019; Wang et al., 2022b), it is difficult to optimize very deep PLMs, especially when shared components are involved. In this section, we introduce a simple yet stable training algorithm for MPO-based PLM and then discuss how it addresses the training instability issue.

### 3.3.1 MPO-based Network Initialization

Existing work has found that parameter initialization is important for training deep models (Huang et al., 2020; Zhang et al., 2019; Wang et al., 2022b), which can help alleviate the training instability. To better optimize the scaling PLMs, we propose a specially designed initialization method based on the above MPO-based architecture.

**Initialization with MPO Decomposition**. Since MPO-based architecture shares global components (*i.e.,* the central tensor) across all layers, our idea is to employ existing well-trained PLMs based on weight sharing for improving parameter initialization. Here, we use the released 24-layer AL-BERT with all the parameters shared across layers. The key idea is to perform MPO decomposition on the parameter matrices of the ALBERT model and obtain the corresponding central and auxiliary tensors. We first divide the model into several groups by structure (embedding, attention, and feed-forward network). Then, for each group, We initialize central tensors by the derived central tensors from the MPO decomposition results of ALBERT. Since they are globally shared, one single copy is only needed for initialization regardless of the layer depth. Next, for *auxiliary tensors*, we directly copy the auxiliary tensors from the MPO decomposition results of ALBERT.

**Scaling the Initialization**. A potential issue is that ALBERT only provides a 24-layer architecture, and such a strategy no longer supports the initialization for an architecture of more than 24 layers (without corresponding auxiliary tensors). As our solution, Inspired by the idea in Wang et al. (2022b) that avoids the exploding update by incorporating an additional scaling coefficient and multiplying the randomly initialized values for the auxiliary tensors (those in higher than 24 layers) with a coefficient of $(2L)^{-\frac{1}{4}}$, where $L$ is the layer number. Then, we present a theoretical analysis of training stability.

### 3.3.2 Theoretical Analysis

To understand the issue of training instability from a theoretical perspective, we consider a Transformer-based model $F(\boldsymbol{x}, \mathbf{W})$ with $\boldsymbol{x}$ and $\mathbf{W}$ as input and parameters, and consider one-step update $\triangle F$[1]. According to Wang et al. (2022b), a large model update ($\triangle F$) at the beginning of training is likely to cause the training instability of deep

---
[1]$\triangle F \triangleq F(\boldsymbol{x}, \mathbf{W} - \eta \frac{\partial}{\partial \mathbf{W}} \mathcal{L}(F(\boldsymbol{x}) - y)) - F(\boldsymbol{x}; \mathbf{W})$.

Transformer models. To mitigate the exploding update problem, the update should be bounded by a constant, *i.e.,* $\|\triangle F\| = \mathcal{O}(1)$. Next, we study how the $\triangle F$ is bounded with the MPOBERT.

**MPO-based Update Bound**. Without loss of generality, we consider a simple case of low-order MPO decomposition: $n = 3$ in Eq. (4). Following the derivation method in Wang et al. (2022b), we simplify the matrices $\mathbf{W}$, $\mathcal{A}_1$, $\mathcal{C}$ and $\mathcal{A}_2$ to scalars $w,u,c,v$, which means the parameter $w_l$ at the $l$-th layer can be decomposed as $w_l = u_l \cdot c_l \cdot v_l$. Based on these notations, we consider $L$-layer transformer-based model $F(x,w)(w = \{w_1, w_2, ..., w_L\})$, where each sub-layer is normalized with Post-LN: $x_{l+1} = LN(x_l + G_l(x_l, w_l))$. Then we can prove $\|\triangle F\|$ satisfies (see Theorem A.1 in the Appendix):

$$\|\triangle F\| \le \sum_{l=1}^{L} (c_1 v_l \|u_l^* - u_l\| + c_1 u_l \|v_l^* - v_l\| + v_l u_l \|c_1^* - c_1\|), \tag{2}$$

The above equation bounds the model update in terms of the central and auxiliary tensors. Since central tensors ($c_l$) can be initialized using the pre-trained weights, we can further simplify the above bound by reducing them. With some derivations (See Corollary A.2 in the Appendix), we can obtain $(v_i^2 + u_i^2)(u_L v_L) = \mathcal{O}(\frac{1}{L})$ in order to guarantee that $\|\triangle F\| = \mathcal{O}(1)$. For simplicity, we set $u_i = v_i = (2L)^{-\frac{1}{4}}$ to bound the magnitude of each update independent of layer number $L$. In the implementation, we first adopt the Xavier method for initialization, and then scale the parameter values with the coefficient of $(2L)^{-\frac{1}{4}}$.

**Comparison**. Previous research has shown that using designed values for random initialization can improve the training of deep models (Huang et al., 2020; Zhang et al., 2019; Wang et al., 2022b). These methods aim to improve the initialization of general transformer-based architectures for training from scratch. As a comparison, we explore the use of pre-trained weights and employ the MPO decomposition results for initialization. In particular, Gong et al. (2019) have demonstrated the effectiveness of stacking pre-trained shallow layers for deep models in accelerating convergence, also showing performance superiority of pre-trained weights over random initialization.

### 3.3.3 Training and Acceleration

To instantiate our approach, we pre-train a 48-layer BERT model (*i.e.,* MPOBERT$_{48}$). For a fair comparison with BERT$_{\text{BASE}}$ and BERT$_{\text{LARGE}}$, we adopt the same pre-training corpus (BOOKCORPUS (Zhu et al., 2015) and English Wikipedia (Devlin et al., 2018)) and pre-training tasks (masked language modeling, and sentence-order prediction). We first perform MPO decomposition on the weights of ALBERT and employ the initialization algorithm in Section 3.3.1 to set the parameter weights. During the training, we need to keep an updated copy of central tensors and auxiliary tensors: we optimize them according to the pre-training tasks in an end-to-end way and combine them to derive the original parameter matrix for forward computation (taking a relatively small cost of parallel matrix multiplication).

Typically, the speed of the pre-training process is affected by three major factors: arithmetic bandwidth, memory bandwidth, or latency. We further utilize a series of efficiency optimization ways to accelerate the pre-training, such as mixed precision training with FP16 (reducing memory and arithmetic bandwidth) and fused implementation of activation and normalization (reducing latency). Finally, we can train the 48-layer MPOBERT at a time cost of 3.8 days (compared with a non-optimized cost of 12.5 days) on our server configuration (8 NVIDIA V100 GPU cards and 32GB memory). More training details are can be found in the experimental setup Section 4.1 and Appendix A.3 (Table 6 and Algorithm 2).

## 4 Experiments

In this section, we first set up the experiments and then evaluate the efficiency of MPOBERT on a variety of tasks with different model settings.

### 4.1 Experimental Setup

**Pre-training Setup**. For the architecture, we denote the number of layers as $L$, the hidden size as $H$, and the number of self-attention heads as $A$. We report results on four model sizes: **MPOBERT$_{12}$** ($L$=12, $H$=768, $A$=12), **MPOBERT$_{48}$** ($L$=48, $H$=1024, $A$=16) and **MPOBERT$_{48+}$** that implement cross-layer parameter sharing in three distinct groups as discussed in subsection 3.2.2. We pre-train all of the models with a batch size of 4096 for $10k$ steps.

| Experiments | MRPC F1 | SST-2 Acc. | CoLA Mcc. | RTE Acc. | STS-B Spear. | QQP F1/Acc. | MNLI Acc. | QNLI Acc. | SQuAD F1 | Avg. | #To (M) |
|---|---|---|---|---|---|---|---|---|---|---|---|
| *Development set* | | | | | | | | | | | |
| **Small Models (#To < 100M)** | | | | | | | | | | | |
| ALBERT$_{12}$ | 89.0 | 90.6 | 53.4 | 71.1 | 88.2 | -/89.1 | 84.5 | 89.4 | 89.3 | 82.7 | 11 |
| ALBERT$_{24}$ | 84.6 | 93.6 | 52.5 | **79.8** | 90.1 | -/88.1 | 85.0 | 91.7 | 90.6 | 84.0 | 18 |
| T5$_{12}$ | 89.2 | **94.7** | 53.5 | 71.7 | 91.2 | -/91.1 | 87.8 | 93.8 | 90.0 | 84.8 | 60 |
| MPOBERT$_{12}$ | 90.3 | 92.3 | 55.2 | 71.8 | 90.5 | -/90.1 | 84.7 | 91.2 | 90.1 | 84.0 | 20 |
| MPOBERT$_{48}$ | **90.8** | 94.7 | **58.3** | 77.3 | 91.4 | -/89.5 | 86.3 | 92.0 | 92.3 | **85.8** | 75 |
| **Base Models (#To > 100M)** | | | | | | | | | | | |
| BERT$_{12}$ | 90.7 | 91.7 | 48.9 | 71.4 | 91.0 | -/90.8 | 83.7 | 89.3 | 88.5 | 82.9 | 110 |
| XLNet$_{12}$ | 85.3 | **94.4** | 49.3 | 63.9 | 85.6 | -/90.7 | **90.9** | 91.8 | 90.2 | 82.5 | 117 |
| RoBERTa$_{12}$ | **91.9** | 92.2 | **59.4** | 72.2 | 89.4 | -/91.2 | 88.0 | 92.7 | 91.2 | 85.4 | 125 |
| BART$_{12}$ | 91.4 | 93.8 | 56.3 | 79.1 | 89.9 | -/90.8 | 86.4 | 92.4 | 91.6 | 82.8 | 140 |
| MPOBERT$_{48+}$ | 89.7 | **94.4** | 57.4 | **79.8** | 91.1 | -/89.3 | 87.1 | 92.4 | 91.4 | **86.0** | 102 |
| *Test set* | | | | | | | | | | | |
| **Small Models (#To < 100M)** | | | | | | | | | | | |
| ALBERT$_{12}$ | 89.2 | 93.2 | 53.6 | 70.2 | 87.3 | 70.3/- | 84.6 | 92.5 | 89.3 | 81.1 | 11 |
| ALBERT$_{24}$ | 88.7 | **94.0** | 51.7 | **73.7** | 86.9 | 69.1/- | 84.9 | **91.8** | 90.6 | 81.2 | 18 |
| MobileBERT$_{24}$♦ | 88.8 | 92.6 | 51.1 | 70.4 | 84.8 | 70.5/- | 83.3 | 91.6 | 90.3 | 80.4 | 25 |
| T5$_{12}$ | 89.7 | 91.8 | 41.0 | 69.9 | 85.6 | 70.0/- | 82.4 | 90.3 | 90.0 | 78.7 | 60 |
| TinyBERT$_{6}$♣ | 87.3 | 93.1 | 51.1 | 70.0 | 83.7 | **71.6/-** | 84.6 | 90.4 | 87.5 | 79.9 | 67 |
| DistilBERT$_{6}$♣ | 86.9 | 92.5 | 49.0 | 58.4 | 81.3 | 70.1/- | 82.6 | 88.9 | 86.2 | 77.3 | 67 |
| MPOBERT$_{12}$ | 89.2 | 91.9 | 52.7 | 70.6 | 87.1 | 69.6/- | 85.0 | 91.0 | 90.1 | 80.8 | 20 |
| MPOBERT$_{48}$ | **90.0** | **94.0** | 55.0 | 74.0 | 88.7 | 71.0/- | 86.5 | 91.8 | 92.3 | 82.6 | 75 |
| **Base Models (#To > 100M)** | | | | | | | | | | | |
| BERT$_{12}$♠ | 88.9 | 93.5 | 52.1 | 66.4 | 85.8 | 71.2/- | 84.6 | 90.5 | 88.5 | 79.1 | 110 |
| XLNet$_{12}$ | 89.2 | 94.3 | 47.3 | 66.5 | 85.4 | 71.9/- | 87.1 | 91.4 | 90.2 | 80.4 | 117 |
| RoBERTa$_{12}$ | 89.9 | 93.2 | **57.9** | 69.9 | 88.3 | **72.5/-** | **87.7** | 92.5 | 91.2 | 82.6 | 125 |
| BART$_{12}$ | 89.9 | 93.7 | 49.6 | 72.6 | 86.9 | 71.7/- | 84.9 | 92.3 | 91.6 | 81.5 | 140 |
| MPOBERT$_{48+}$ | **89.9** | **94.5** | 56.0 | **74.5** | 88.4 | 70.5/- | 86.5 | 92.6 | 91.4 | 82.7 | 102 |

Table 1: Performance comparison of different models on natural language understanding tasks (in percent). "# To (M)" denote the number (in millions) of total parameters. We compare MPOBERT with PLMs (*i.e.,* BERT and ALBERT) and Parameter-efficient Transformers (*i.e.,* MobileBERT, TinyBERT and DistilBERT), respectively. The best and the second-best performance in each task are highlighted in bold and underlined. ♦: Experimental results by Sun et al. (2020b). ♣: Experimental results by Jiao et al. (2019). ♠: Experimental results by Devlin et al. (2018).

**Fine-tuning Datasets.** To evaluate the performance of our model, we conduct experiments on the GLUE (Wang et al., 2018) and SQuAD v1.1 (Rajpurkar et al., 2016) benchmarks. Since fine-tuning is typically fast, we run an exhaustive parameter search and choose the model that performs best on the development set to make predictions on the test set. We include the details in the Appendix(see Appendix A.4.1 for the datasets and Appendix A.4.2 for evaluation metrics)

**Baseline Models.** We compare our proposed MPOBERT to the existing competitive deep PLMs and parameter-efficient models. In order to make fair comparisons, we divide the models into three major categories based on their model sizes:

• Small models (#To <100M). ALBERT$_{12}$ (Lan et al., 2019) is the most representative PLM that achieves competitive results with only 11M. In addition, we consider PLMs (T5$_{12}$) and three

compressed models that have similar parameters, namely MobileBERT (Sun et al., 2020b), DistilBERT (Sanh et al., 2019) and TinyBERT (Jiao et al., 2019). We compare these compressed models to show the benefit of scaling to deeper models over compressing large models to small variants.

• Base models (#To > 100M). We compare with BERT$_{12}$, XLNet$_{12}$, RoBERTa$_{12}$ and BART$_{12}$ for this category. Note that we only include the base variants that have similar model sizes in order to make a fair comparison.

More details about the experiments are described in Appendix A.4.

## 4.2 Main Results

**Fully-supervised setting.** We present the results of MPOBERT and other baseline models on GLUE and Squad for fine-tuning in Table 1.

Firstly, we evaluate MPOBERT's performance

in comparison to other models with similar numbers of parameters. In particular, for small models, MPOBERT$_{48}$ outperforms the best baseline models and achieves substantial improvements on both the development set (85.8 *v.s.* 84.8 for T5$_{12}$) and test sets (82.6 *v.s.* 81.2 for ALBERT$_{24}$). This highlights the benefits of increased capacity from layer-specific parameters (*i.e.,* the auxiliary tensors and layer-specific adapters) in MPOBERT. Furthermore, for small and base models, 48-layer MPOBERT consistently achieves better results than all parameter-efficient models, while also achieving comparable results to other 12-layer PLMs with a reduced number of parameters. This demonstrates the significant benefits of scaling along the model depth with layer-specific parameters in MPOBERT.

Secondly, we assess MPOBERT's parameter efficiency by comparing it to other PLMs within the same model depth. For instance, when considering models with $L$=12 layers, MPOBERT achieves comparable results or even outperforms (+1.7 for BERT$_{12}$ and +0.4 for XLNet$_{12}$) PLMs while having fewer parameters. This further highlights the advantages of MPOBERT's parameter-efficient approach in constructing deep models.

**Multitask Fine-tuning Setting**. To demonstrate the effectiveness of our proposed parameter-sharing model in learning shared representations across multiple tasks, we fine-tune MPOBERT, BERT and ALBERT on the multitask GLUE benchmark and report the results in Table 2. Specifically, we design two groups of experiments. (1) Deep vs. shallow models. Comparing with BERT$_{12}$, MPOBERT$_{48}$ has much deeper Transformer layers but still fewer total parameters (*i.e.,* 75M vs. 110M). We find that MPOBERT$_{48}$ achieves 1.4 points higher on average GLUE score than BERT$_{12}$. (2) Central tensors sharing vs. all weight sharing. Comparing with ALBERT$_{12}$, MPOBERT$_{12}$ only shares part of weights, *i.e.,* central tensors, while ALBERT$_{12}$ shares all of the weights. We find that sharing central tensors may effectively improve the average results than sharing all weights (82.0 *v.s.* 81.4 for MRPC).

**Few-shot Learning Setting**. We evaluate the performance of our proposed model, MPOBERT, in few-shot learning setting (Huang et al., 2022) on two tasks, SST-2 and MNLI, using a limited number of labeled examples. Results in Table 3 show

| Datasets | B$_{12}$ | M$_{48}$ | M$_{12}$ | A$_{12}$ |
|---|---|---|---|---|
| MNLI (Acc.) | 83.9 | 85.4 | 82.8 | 82.7 |
| QNLI (Acc.) | 90.8 | 91.1 | 90.0 | 89.4 |
| SST-2 (Acc.) | 91.7 | 93.0 | 90.9 | 90.6 |
| RTE (Acc.) | 81.2 | 82.0 | 79.8 | 79.1 |
| QQP (Acc.) | 91.2 | 87.6 | 90.4 | 89.7 |
| CoLA (Mcc.) | 53.6 | 54.9 | 45.0 | 35.9 |
| MRPC (F1) | 84.2 | 91.8 | 89.9 | 89.2 |
| STS-B (Spear.) | 87.4 | 89.0 | 86.9 | 87.5 |
| Avg. | 83.0 | 84.4 | 82.0 | 80.5 |
| #To (M) | 110 | 75 | 20 | 11 |

Table 2: Performance of multi-task learning on GLUE benchmark obtained by fine-tuning BERT$_{12}$ (B$_{12}$), MPOBERT$_{48}$ (M$_{48}$), MPOBERT$_{12}$ (M$_{12}$) and ALBERT$_{12}$ (A$_{12}$) (in percent).

| | SST-2 | | | MNLI | | |
|---|---|---|---|---|---|---|
| Shots (K) | 10 | 20 | 30 | 10 | 20 | 30 |
| BERT$_{12}$ | 54.8 | 59.7 | 61.6 | **37.0** | 35.6 | 35.7 |
| ALBERT$_{12}$ | 56.7 | 59.3 | 60.0 | 36.3 | 35.6 | 36.5 |
| MPOBERT$_{12}$ | **58.9** | **65.4** | **64.6** | 36.7 | **36.7** | **37.1** |

Table 3: Comparison of few-shot performance.

that MPOBERT outperforms BERT, which suffers from over-fitting, and ALBERT, which does not benefit from its reduced number of parameters. These results further demonstrate the superiority of our proposed model in exploiting the potential of large model capacity under limited data scenarios.

### 4.3 Detailed Analysis

**Analysis of Initialization Methods**. This experiment aims to exclude the effect of initialized pre-trained weights on fine-tuning results. We plot the performance of the model on SST-2 *w.r.t* training steps. In particular, we compare the performance of MPOBERT using different initialization methods (Xavier in Fig. 3(a) and decomposed weights of ALBERT in Fig. 3(b)) for pre-training. The results demonstrate that pre-training MPOBERT from scratch requires around 50$k$ steps to achieve performance comparable to BERT$_{\text{BASE}}$, while initializing with the decomposed weights of ALBERT significantly accelerates convergence and leads to obvious improvements within the first 10$k$ training steps. In contrast, the gains from continual pre-training for ALBERT are negligible. These results provide assurance that the improvements observed in MPOBERT are not solely attributed to the use of initialized pre-trained weights.

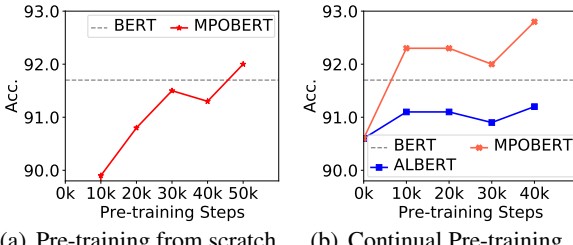

(a) Pre-training from scratch  (b) Continual Pre-training

Figure 3: Comparison of the SST-2 accuracy achieved through pre-training from scratch and pre-training with the initialization of decomposed ALBERT weights.

| Experiment | SST-2 | RTE | MRPC | #To (M) |
|---|---|---|---|---|
| MPOBERT$_{12}$ | 92.8 | 72.9 | 91.8 | 20.0 |
| w/o Adapter | 92.3 | 71.8 | 90.3 | 19.4 |
| w/o PS | 91.4 | 67.9 | 85.8 | 11.9 |

Table 4: Ablation study on the SST-2, RTE, and MRPC datasets (in percent).

| Rank | SST-2 | RTE | MRPC | #To (M) |
|---|---|---|---|---|
| 4 | 91.9 | 69.7 | 88.2 | 19.7 |
| 8 | 92.8 | 72.9 | 91.8 | 20.0 |
| 64 | 91.6 | 69.3 | 88.1 | 24.3 |

Table 5: Comparison of different adapter ranks on three GLUE tasks (in percent). "Rank" denotes the adapter rank in MPOBERT.

also observe a decrease in the performance of the variant with adapter rank 64. This illustrates that further increasing the rank may increase the risk of over-fitting in fine-tuning process. Therefore, we set a rank of 8 for MPOBERT in the main results.

**Analysis of Linguistic Patterns**. To investigate the linguistic patterns captured by MPOBERT, BERT, and ALBERT, we conduct a suite of probing tasks, following the methodology of Tenney et al. (2019). These tasks are designed to evaluate the encoding of surface, syntactic, and semantic information in the models' representations. The results, shown in Fig. 4, reveal that BERT encodes more local syntax in lower layers and more complex semantics in higher layers, while ALBERT does not exhibit such a clear trend. However, MPOBERT exhibits similar layer-wise behavior to BERT in some tasks (*i.e.,* task 0,2,4), and improved results in lower layers for others (*i.e.,* task 3) which is similar to ALBERT. The result demonstrates that MPOBERT captures linguistic information differently than other models, and its layer-wise parameters play an important role in this difference.

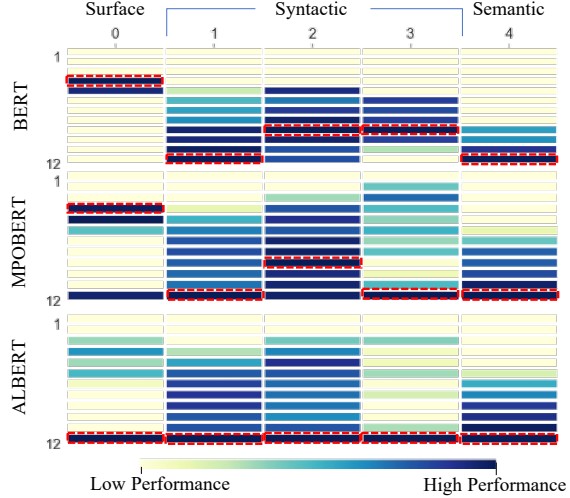

Figure 4: A visualization of layer-wise linguistic patterns. Each column represents a probing task, and each row represents a Transformer layer. The red dashed box indicates the layer that performs best.

**Ablation Analysis**. To assess the individual impact of the components in our MPOBERT model, we conduct an ablation study by removing either the layer-specific adapter or the cross-layer parameter-sharing strategy. The results, displayed in Table 4, indicate that the removal of either component results in a decrease in the model's performance, highlighting the importance of both components in our proposed strategy. While the results also indicate that cross-layer parameter sharing plays a more crucial role in the model's performance.

**Performance Comparison *w.r.t* Adapter Rank**. To compare the impact of the adapter rank in layer-specific adapters on MPOBERT's performance, we trained MPOBERT with different ranks (4,8 and 64) and evaluate the model on downstream tasks in Table 5. The results demonstrate that a rank of 8 is sufficient for MPOBERT, which further shows the necessity of layer-specific adapters. However, we

## 5   Conclusion

We develop MPOBERT, a parameter-efficient pre-trained language model that allows for the efficient scaling of deep models without the need for additional parameters or computational resources. We achieve this by introducing an MPO-based Transformer layer and sharing the central tensors across layers. During training, we propose initialization methods for the central and auxiliary tensors, which are based on theoretical analysis to address training

stability issues. The effectiveness of MPOBERT is demonstrated through various experiments, such as supervised, multitasking, and few-shot where it consistently outperforms other competing models.

## Limitations

The results presented in our study are limited by some natural language processing tasks and datasets that are evaluated, and further research is needed to fully understand the interpretability and robustness of our MPOBERT models. Additionally, there is subjectivity in the selection of downstream tasks and datasets, despite our use of widely recognized categorizations from the literature. Furthermore, the computational constraints limited our ability to study the scaling performance of the MPOBERT model at deeper depths such as 96 layers or more. This is an area for future research.

## Ethics Statement

The use of a large corpus for training large language models may raise ethical concerns, particularly regarding the potential for bias in the data. In our study, we take precautions to minimize this issue by utilizing only standard training data sources, such as BOOKCORPUS and Wikipedia, which are widely used in language model training (Devlin et al., 2018; Lan et al., 2019). However, it is important to note that when applying our method to other datasets, the potential bias must be carefully considered and addressed. Further investigation and attention should be given to this issue in future studies.

## Acknowledgments

This work was partially supported by National Natural Science Foundation of China under Grants No. 62206299 and 62222215, Beijing Natural Science Foundation under grant No. L233008, Beijing Outstanding Young Scientist Program under Grant No. BJJWZYJH012019100020098 and CCF-Zhipu AI Large Model Fund. Xin Zhao is the corresponding author.

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

# A  Appendix

## A.1  Matrix Product Operators

Formally, given a weight matrix $\mathbf{W} \in \mathbb{R}^{I \times J}$, we can factorize the two dimensions into a product of natural numbers, and reshape it into an $n$-dimension tensor $\mathbf{W}_{i_1,\ldots,i_n,j_1,\ldots,j_n}$, which satisfies:

$$\prod_{k=1}^{n} i_k = I, \quad \prod_{k=1}^{n} j_k = J. \quad (3)$$

This decomposition can be written as:

$$\mathbf{W}_{i_1,\ldots,i_n,j_1,\ldots,j_n} = \mathcal{T}^{(1)}[i_1,j_1]\cdots\mathcal{T}^{(n)}[i_n,j_n], \quad (4)$$

where the $\mathcal{T}^{(k)}[i_k,j_k]$ is a 4-dimensional tensor with size $d_{k-1} \times i_k \times j_k \times d_k$ in which $d_k$ is a bond dimension linking $T^{(k)}$ and $T^{(k+1)}$ with $d_0 = d_n = 1$. with size $d_{k-1} \times i_k \times j_k \times d_k$ in which $\prod_{k=1}^{n} i_k = I, \prod_{k=1}^{n} j_k = J$ and $d_0 = d_n = 1$. This bond dimension indicates the associative strength between two adjacent tensors. For clarity, we can rewrite the decomposition results as central tensor $\mathcal{C}$ and auxiliary tensors $\{\mathcal{A}_i\}_{i=1}^{n-1}$. As an important merit, such a decomposition can effectively reorganize and aggregate the information of the matrix (Gao et al., 2020a): central tensor $\mathcal{C}$ can encode the essential information from the original matrix, while auxiliary tensors $\{\mathcal{A}_i\}_{i=1}^{n-1}$ serve as its complement to precisely reconstruct the matrix.

The $k$-th order and $k \in \{1,\ldots,D\}$. The bond dimension $d_k$ is defined by:

$$d_k = \min\left(\prod_{m=1}^{k} i_m \times j_m, \prod_{m=k+1}^{n} i_m \times j_m\right). \quad (5)$$

From Eq. (5), we can see that is going to be large in the middle and small on both sides. Algorithm 1 presents a thorough algorithm for MPO decomposition.

---

**Algorithm 1** MPO decomposition procedure.

**Require:** matrix $\mathbf{W} \in \mathbb{R}^{I \times J}$, the number of local tensor $m$
**Output** : local tensor set $\{\mathcal{T}^{(k)}\}_{k=1}^{m}$
1: **for** $k = 1,\ldots,m-1$ **do**
2: $\quad \mathbf{W}[d_{k-1} \times i_k \times j_k, -1] \leftarrow \text{Reshape}(\mathbf{W}[I,J])$
3: $\quad \mathbf{U}\lambda\mathbf{V}^{\top} \leftarrow \text{SVD}(\mathbf{W})$
4: $\quad \mathcal{T}^{(k)}[d_{k-1}, i_k, j_k, d_k] \leftarrow \text{Reshape}(\mathbf{U})$
5: $\quad$ Calculate $\mathbf{W} = \lambda\mathbf{V}^{\top}$
6: **end for**
7: Let $\mathcal{T}^{(m)} \leftarrow \mathbf{W}$
8: Normalization
9: **return** local tensor set $\{\mathcal{T}^{(k)}\}_{k=1}^{m}$

---

The MPO representation of $\mathbf{W}$ is obtained by factorizing it into a sequential product of local tensors.

## A.2  Proofs

**Notations.** We denote $\mathcal{L}(\cdot)$ as the loss function. $LN(x)$ as the standard layer normalization with scale $\gamma = 1$ and bias $\beta = 0$. Let $\mathcal{O}(\cdot)$ denote standard Big-O notation that suppresses multiplicative constants. $\overset{\Theta}{=}$ stands for equal bound of magnitude. We aim to study the magnitude of the model updates. We define the model update as $\|\triangle F\|$.

**Definition** $F(x,\theta)$ is updated by $\Theta(\eta)$ per SGD step after initialization as $\eta \rightarrow 0$. That is, $\|\triangle F(x)\| = \Theta(\eta)$ where $\triangle F(x)$ can be calculated through $F(x, \theta - \eta\frac{\partial}{\partial\theta}\mathcal{L}(F(x)-y)) - F(x;\theta)$.

**Theorem A.1** *Given an N-layer transformer-based model $F(x,\theta)(\theta = \{\theta_1,\theta_2,\ldots,\theta_N\})$, where $\theta_l$ denotes the parameters in $l$-th layer and each sub-layer is normalized with Post-LN: $x_{l+1} = LN(x_l + G_l(x_l,\theta_l))$. In MPOBERT, $\theta_l$ is decomposed by MPO to local tensors: $\theta_l = u_l \cdot c_l \cdot v_l$, and we share $\{c_i\}_{i=1}^{N}$ across $N$ layers: $c_l = c_1, l = 1,2,\cdots,N$. Then $\|\triangle F\|$ satisfies:*

$$\|\triangle F\| \leq \sum_{i=1}^{N}(c_1 v_i \|u_i^* - u_i\| + c_1 u_i \|v_i^* - v_i\| \\ + v_i u_i \|c_1^* - c_1\|) \quad (6)$$

*Proof.* We follow (Zhang et al., 2019) and make the following assumptions to simplify the derivations:

1. Hidden dimension $d$ equals to 1;

2. $var(x + G_l(x)) \overset{\Theta}{=} var(x) + var(G_l(x))$;

3. All relevant weights $\theta$ are positive with magnitude less than 1.

Given Assumption 1, if $G_l(x)$ is MLP with the weight $\theta_l$, then $G_l(x) \overset{\Theta}{=} \theta_l x$. With assumption 2, we have:

$$x_{l+1} = f_l(x_l,\theta_l) = \frac{x + G_l(x)}{\sqrt{Var(x + G_l(x))}} \quad (7)$$

$$\overset{\Theta}{=} \frac{1 + \theta_l}{\sqrt{1 + \theta_l^2}}x_l, \quad (8)$$

Then, with Taylor expansion, the model update

$\|\triangle F\|$ satisfies:

$$
\begin{aligned}
\|\triangle F\| &= \|F(x, \theta^*) - F(x, \theta)\| \\
&= \|x_{N+1}^* - x_{N+1}\| \\
&= \|f(x_N^*, \theta_N^*) - f(x_N, \theta_N)\| \\
&= \|f(x_N^*, U_N^*, C_N^*, V_N^*) \\
&\quad - f(x_N, U_N, C_N, V_N)\| \\
&\approx \left\| \frac{\partial f}{\partial x}(x_N^* - x_N) \right. \\
&\quad + \frac{\partial f}{\partial \theta}\frac{\partial \theta}{\partial U_N}(U_N^* - U_N)^T \\
&\quad + \frac{\partial f}{\partial \theta}\frac{\partial \theta}{\partial C_N}(C_N^* - C_N)^T \\
&\quad + \left. \frac{\partial f}{\partial \theta}\frac{\partial \theta}{\partial V_N}(V_N^* - V_N)^T \right\|
\end{aligned} \tag{9}
$$

With Eq. (8), the magnitude of $\frac{\partial f_l}{\partial x}$ and $\frac{\partial f_l}{\partial \theta}$ is bounded by:

$$
\frac{\partial f_l}{\partial x} \overset{\Theta}{=} \frac{1 + \theta_l}{\sqrt{1 + \theta_l^2}} \tag{10}
$$

$$
\frac{\partial f_l}{\partial \theta_l} \overset{\Theta}{=} \frac{1 - \theta_l}{(1 + \theta_l^2)^{\frac{3}{2}}} x_l \tag{11}
$$

Since we apply MPO decomposition to $\theta_l$, we get:

$$
\theta_l = U_l \cdot C_l \cdot V_l \tag{12}
$$

For simplicity, we reduce the matrices $U, C, V$ to the scalars $u, c, v$. Thus with Assumption 3, Eq. (9) is reformulated as: Finally, with Assumption 3 we have:

$$
\begin{aligned}
\|\triangle F\| &= \|x_{N+1}^* - x_{N+1}\| \tag{13} \\
&\leq \sum_{i=1}^N \frac{1 - u_i c_1 v_i}{(1 + u_i^2 c_1^2 v_i^2)^{\frac{3}{2}}} (c_1 v_i \|u_i^* - u_i\| \\
&\quad + c_1 u_i \|v_i^* - v_i\|) + v_i u_i \|c_1^* - c_1\|) \\
&\approx \sum_{i=1}^N (c_1 v_i \|u_i^* - u_i\| + c_1 u_i \|v_i^* - v_i\| \\
&\quad + v_i u_i \|c_1^* - c_1\|)
\end{aligned} \tag{14}
$$

$\square$

**Corollary A.2** *Given that we initialise $c_1$ in MPOBERT with well-trained weights, it is reasonable to assume that updates of $c_1$ are well-bounded. Then $\triangle F$ satisfies $\|\triangle F\| = \mathcal{O}(1)$ when for all $i = 1, \cdots, N$:*

$$
(v_i^2 + u_i^2)(u_N v_N) = \mathcal{O}(\frac{1}{N}) \tag{15}
$$

$Proof.$ For an $N$-layer MPOBERT, we have:

$$
\|\triangle F\| \leq \sum_{i=1}^N (v_i \|u_i^* - u_i\| + u_i \|v_i^* - v_i\|) \tag{16}
$$

$$
\begin{aligned}
\leq \eta \sum_{i=1}^N (&v_i \left\| \frac{\partial \mathcal{L}}{\partial F} \right\| \cdot \left\| \frac{\partial F}{\partial \theta_i} \right\| \cdot \left\| \frac{\partial \theta_i}{\partial u_i} \right\| \\
&+ u_i \left\| \frac{\partial \mathcal{L}}{\partial F} \right\| \cdot \left\| \frac{\partial F}{\partial \theta_i} \right\| \cdot \left\| \frac{\partial \theta_i}{\partial v_i} \right\|)
\end{aligned} \tag{17}
$$

By assumption $\left\| \frac{\partial \mathcal{L}}{\partial F} \right\| = \mathcal{O}(1)$ and $\left\| \frac{\partial F}{\partial \theta_i} \right\| \leq \left\| \frac{\partial F}{\partial \theta_N} \right\| \overset{\Theta}{=} \|\theta_N\|$, we achieve:

$$
\begin{aligned}
&\eta \sum_{i=1}^N (v_i \left\| \frac{\partial \mathcal{L}}{\partial F} \right\| \cdot \left\| \frac{\partial F}{\partial \theta_i} \right\| \cdot \left\| \frac{\partial \theta_i}{\partial u_i} \right\| \\
&\quad + u_i \left\| \frac{\partial \mathcal{L}}{\partial F} \right\| \cdot \left\| \frac{\partial F}{\partial \theta_i} \right\| \cdot \left\| \frac{\partial \theta_i}{\partial v_i} \right\|) \\
&= \eta \sum_{i=1}^N (v_i^2 u_N v_N + u_i^2 u_N v_N) \\
&= \mathcal{O}(\sum_{i=1}^N (v_i^2 + u_i^2)(u_N v_N)) = \mathcal{O}(1),
\end{aligned} \tag{18} \tag{19}
$$

Finally, we achieve:

$$
(v_i^2 + u_i^2)(u_N v_N) = \mathcal{O}(\frac{1}{N}) \tag{20}
$$

Due to symmetry, we set $u_i = u$, $v_i = v$. Thus, from A.2, we set $u = v = (2N)^{-\frac{1}{4}}$ to achieve to bound the magnitudes of each update to be independent of model depth $N$, *i.e.,* $\|\triangle F\| = \mathcal{O}(1)$.

$\square$

### A.2.1 Details of Training
### A.3 Training Details

Here we describe the details of the pre-training process in Algorithm 2. For pre-training, we tune the learning rate in the range of $[1.0 \times 10^{-5}, 1.0 \times 10^{-6}]$ and use the LAMB optimizer (You et al., 2020). Since fine-tuning is typically fast, we run an exhaustive parameter search (*i.e.,* learning rate in the range of $[2.0 \times 10^{-4}, 2.0 \times 10^{-6}]$, batch size in $\{8, 16, 32\}$) and choose the model that performs best on the development set to make predictions on the test set.

### A.3.1 Details of Training Configurations

In this part, we list the training configurations of MPOBERT and other representative PLMs in Table 6.

| Models | #To (M) | Depth | Samples | Training time | GLUR Dev. | GLUE Test |
|--------|---------|-------|---------|---------------|-----------|-----------|
| T5$_{11B}$ | 11000 | 24 | - | - | - | 89.0 |
| T5$_{BASE}$ | 220 | 24 | $128 \times 524k$ | 16 TPU v3 1 Day (t5-base) | 84.1 | 82.5 |
| BERT$_{LARGE}$ | 330 | 24 | $256 \times 1000k$ | 16 Cloud TPUs 4 Days | 84.1 | 81.6 |
| ALBERT$_{XXLARGE}$ | 235 | 1 | $4096 \times 1.5M$ | TPU v3 16 Days | 90.0 | - |
| BART$_{LARGE}$ | 407 | 24 | $8000 \times 500k$ | - | 88.8 | - |
| RoBERTa$_{LARGE}$ | 355 | 24 | $8000 \times 500k$ | 1024 V100 GPUs 1 Day | 88.9 | - |
| XLNet$_{LARGE}$ | 361 | 24 | $8192 \times 500k$ | 512 TPU v3 5.5 Days | 87.4 | - |
| MPOBERT$_{48+}$ | 102 | 48 | $4096 \times 10k$ | 8 V100 GPUs 3.8 Days | 85.6 | 81.7 |

Table 6: Comparison with the strongest variants of popular PLMs. Since T5$_{11B}$ has far more parameters than other candidates, it's reasonable to use T5$_{base}$ for a fair comparison.

---

**Algorithm 2** The MPOBERT training procedure.

---

**Require:** $\mathbf{W}^{(l)}$: Weight matrix of $l$-th layer in MPOBERT. $\mathbf{W}_A^{(0)}$: Pre-trained weight matrix in ALBERT. $\mathbf{U}^{(l)}$ and $\mathbf{D}^{(l)}$: Matrices in low-rank adapter. $\eta$: Learning rate. $\mathcal{L}$: Stochastic objection function. $L$: Model layers number. (MPO decomposition)
1: $\{\mathcal{A}_1^{(l)}, \mathcal{A}_2^{(l)}, \mathcal{C}^{(l)}, \mathcal{A}_3^{(l)}, \mathcal{A}_4^{(l)}\} \leftarrow \text{MPO}(\mathbf{W}^{(l)})$
2: $\{\mathcal{A}_1^{(0)}, \mathcal{A}_2^{(0)}, \mathcal{C}^{(0)}, \mathcal{A}_3^{(0)}, \mathcal{A}_4^{(0)}\} \leftarrow \text{MPO}(\mathbf{W}_A^{(0)})$ (Initialization Procedure)
3: **for** $0 < l \leq 24$ **do**
4: $\quad \mathcal{C}^{(l)} \leftarrow \mathcal{C}^{(0)}, \{\mathcal{A}_j^{(l)}\}_{j=1}^4 \leftarrow \{\mathcal{A}_j^{(0)}\}_{j=1}^4$
5: **end for**
6: **for** $24 < l \leq L$ **do**
7: $\quad \mathcal{C}^{(l)} \leftarrow \mathcal{C}^{(0)}, \{\mathcal{A}_j^{(l)}\}_{j=1}^4 \leftarrow \{(2L)^{-\frac{1}{4}} \mathcal{A}_j^{(0)}\}_{j=1}^4$
8: **end for**
9: $\mathbf{U}^{(l)} \leftarrow 0, \mathbf{D}^{(l)} \leftarrow \mathcal{N}(0, \sigma^2)$
10: $\mathbf{W}^{(l)} = \mathcal{A}_1^{(l)} \mathcal{A}_2^{(l)} \mathcal{C}^{(l)} \mathcal{A}_3^{(l)} \mathcal{A}_4^{(l)} + \mathbf{W}_{Adapter}^{(l)}$ (Training procedure with mixed precision and fused implementation techniques.)
11: **while** not converged **do**
12: $\quad t \leftarrow t + 1$
13: $\quad g_t \leftarrow \frac{\partial \mathcal{L}(\mathbf{W}_t^{(l)})}{\partial(\mathbf{W}_t^{(l)})}$
14: $\quad \mathbf{W}_t^{(l)} \leftarrow \mathbf{W}_{t-1}^{(l)} - \eta \cdot g_t$
15: **end while**
16: **return** Converged model

---

## A.4 Experimental Details

### A.4.1 Details of Fine-tuning Datasets

GLUE benchmark covers multiple datasets (MNLI, QNLI, QQP, CoLA, RTE, MRPC, SST-2) [2]. The

[2] In line with Raffel et al. (2020), we do not test WNLI due to its adversarial character with respect to the training set.

SQuAD is a collection of $100k$ crowd-sourced question/answer pairs. Given a question and a passage, the task is to predict the answer text span in the passage.

### A.4.2 Details of Evaluation Metrics

Following Gao et al. (2022), we employ Matthew's correlation for CoLA, Spearman for STS-B, F1 for MRPC, and accuracy for the remaining tasks as the metrics for the GLUE benchmark. We compute and present the average scores across all test samples for each of the aforementioned metrics.

## A.5 Analysis of Inference Speed

In this part, we provide FLOPs and inference costs for MPOBERT-48, MPOBERT-12 and BERT-12. Note that we have primarily focused on the FLOPs within the Transformer layer. By conducting multiple experiments and averaging the results, we report the results in Table 7. (1) MPOBERT-12 exhibits higher FLOPs than BERT-12. Notably, owing to MPOBERT-12's significantly reduced parameter count in comparison to BERT-12 (20M vs. 110M), it becomes more feasible to increase the inference batch size within a similar GPU memory (from 4x to 5x). (2) In the case of MPOBERT-48 compared to BERT-12, there is a higher increase in FLOPs. The added computational load originates from increased hyperparameters of layers "L" (48 vs. 12) and hidden dimension size "H" (1024 vs.

Table 7: Model Performance Comparison

| Model | GLUE | GFLOPs | Inference |
|-------|------|--------|-----------|
| BERT-12 | 79.1 | 96.64 | 1.00x |
| MPOBERT-12 | 80.8 | 139.80 | 0.94x |
| MPOBERT-48 | 82.6 | 670.01 | 5.88x |

768). The remarkable enhancement in the performance of MPOBERT-48 is clearly demonstrated by the results of the GLUE Test (82.6 vs. 79.1), underscoring the value of the computational load associated with this part. As a result, we recommend prioritizing MPOBERT-48 in scenarios where performance takes precedence.

It is important to highlight that MPOBERT holds the promise of substantially enhancing inference speed. This is mainly attributed to the fact that the speed of inference is typically constrained by memory bandwidth, which, in turn, is restricted by the available memory capacity. To elucidate this point, envision a situation where the model weights utilized in matrix multiplications are stored in a smaller yet high-bandwidth memory. In such cases, there exists a notable potential for achieving a significant speed boost, particularly when compared with the situation where these weights must be fetched from a larger memory characterized by lower bandwidth.