# OpenReview forum: "Enhancing Scalability of Pre-trained Language Models via Efficient Parameter Sharing"
_EMNLP/2023/Conference — EMNLP 2023 Findings_

### Official Review · Reviewer_M4qE · 2023-07-31

**Soundness:** 4

**Excitement:**

3: Ambivalent: It has merits (e.g., it reports state-of-the-art results, the idea is nice), but there are key weaknesses (e.g., it describes incremental work), and it can significantly benefit from another round of revision. However, I won't object to accepting it if my co-reviewers champion it.

**Missing References:**

The authors should include the paper [1], which is also a MPO-based architecture with the parameter sharing strategy. And discuss the difference in related work. Since the authors argue that their paper has a difference focus from existing work, while [1] optimizes the deep MPO-based architecture with the parameter sharing strategy, and explore the use of well-trained PLMs for initialization.
[1] Exploring extreme parameter compression for pre-trained language models, ICLR 2022

**Paper Topic And Main Contributions:**

This paper presents a parameter-efficient approach, utilizing a matrix product operator (MPO) to scale pre-trained language models (PLMs) to a deep structure. The models are parameterized into shared local tensors and layer specific tensors. To train such models, the authors introduce an MPO-based initialization method, leveraging the MPO decomposition results of ALBERT. Extensive experiments under fully-supervised setting, multitask fine-tuning and few-shot learning settings show the effectiveness of the propose methods.

**Questions For The Authors:**

- Given that the parameter matrices exhibit different shapes, it is imperative to elucidate how the rank for the shared and layer-specific tensors is determined.
- The division of groups in MPOBERT+ remains unclear in the paper, also leaving ambiguity surrounding the differentiation between MPOBERT and MPOBERT+. Is the key distinction solely related to the sharing granularity (layer vs. group)?
- How about the inference speed of MPOBERT compared with models of the same size? Considering that ALBERT's FLOPs remain nearly identical to the original BERT model, the inference speed for MPOBERT48 maybe much more slower than BERT12.
- Does the training speed in table 6 for MPOBERT48+ benefit from initialization  from BERT?

**Reasons To Accept:**

- The authors systematically conduct experiments across various settings, demonstrating a thorough and comprehensive evaluation.
- Theoretical analysis is offered by the authors for initialization.
- The analysis of linguistic pattern is interesting.

**Reasons To Reject:**

- The clarity of the proposed methods is lacking, and essential details necessary for result reproduction are missing.

- Regarding the training of such a model, it appears that the inference speed may be slow, raising potential concerns.

**Reproducibility:**

3: Could reproduce the results with some difficulty. The settings of parameters are underspecified or subjectively determined; the training/evaluation data are not widely available.

**Reviewer Confidence:**

4: Quite sure. I tried to check the important points carefully. It's unlikely, though conceivable, that I missed something that should affect my ratings.

---

> ### Author Rebuttal · Authors · 2023-08-29
>
> We deeply appreciate your perceptive insights, and our detailed responses to your concerns are outlined below. If you have any additional inquiries, please feel free to let us know.
>
> R1Q1:
> We appreciate your insightful feedback. In the MPO decomposition, we note that the central tensor (i.e., the 3rd tensor when we have 5 decomposed tensors) have the most parameters, leading to the choice of central tensors for shared tensors. Empirical validation of rank determination for layer-specific tensors showed consistent performance across various ranks (rank=4, 8, 64 in Table-5). Considering the balance between efficiency and performance, a rank of r=8 is generally a favorable choice. We acknowledge the importance of clarifying these details and intend to incorporate this discussion into the main text to provide a more comprehensive understanding.
>
> R1Q2：
> Your observation is accurate. The fundamental difference between MPOBERT and MPOBERT+ primarily lies in the sharing granularity. MPOBERT+ introduces parameter sharing at the more refined "group" level, thus intending to enhance both the parameter count and the corresponding expressive capacity of the model which potentially leads to enhanced feature extraction and improved overall model performance.
>
> R2Q3：
> In addressing the concern regarding real-time inference latency, we would like to emphasize that the MPO-based Transformer's decomposed tensors undergo a one-time reconstruction into matrices before inference begins. This reconstruction step is incurred only once, and as a result, subsequent inference processes exhibit latency comparable to that of the standard Transformer. Importantly, this approach eliminates the need for additional computations during inference, thereby effectively maintaining efficiency without introducing any overhead.
>
> Q4：
> I guess you are inquiring whether the training speed of MPOBERT_48+ benefits from the initialization method mentioned in our Section 3.3 (ALBERT initialization). Indeed, it does. Referring to the response provided by Reviewer-4FdP (R1), both MPOBERT_48 and MPOBERT_48+ converge to lower loss values quickly. This rapid convergence can be attributed to the favorable initial values that facilitate more stable optimization during the early stages.
>
> Missing reference:
> Thank you for your careful reading and we will cite this work and discuss it.
> The mentioned paper introduces a tensor decomposition technique that relies on the assistance of pre-existing models for knowledge distillation. However, it's important to note that our work revolves around creating novel, deeper models without the reliance on existing larger models for distillation. In this context, relying solely on tensor decomposition is insufficient. Our approach is further enriched by the incorporation of a theoretically guided initialization method to address the training instability issue. These aspects, which distinguish our work, will be thoroughly elaborated upon in our paper.

---

### Official Review · Reviewer_v9WE · 2023-08-04

**Typos Grammar Style And Presentation Improvements:** 1. There seem to be some labeling err…
**Soundness:** 4

**Excitement:**

4: Strong: This paper deepens the understanding of some phenomenon or lowers the barriers to an existing research direction.

**Paper Topic And Main Contributions:**

This paper introduces a highly parameter-efficient approach.
1. The core contribution involves a parameter-sharing architecture based on the Matrix Product Operator (MPO) for deep Transformer networks. To train such deep architectures, an MPO-based initialization method is proposed, leveraging the MPO decomposition results of the ALBERT model. This approach shares the central tensors of parameter matrices across all layers to reduce model size.
2. To maintain layer adaptability, a low-rank adapter is introduced.
3. Theoretical analysis is employed to demonstrate that the proposed MPOBERT can accommodate deeper networks and address training instability issues.
4. The authors conducted thorough experiments.

**Reasons To Accept:**

1. The paper offers a novel perspective on the conventional tensor-train decomposition, introducing a fresh parameter-sharing approach that can be applied to deep models. Ample theoretical derivation supports its stability in training deep architectures, thereby enhancing the method's interpretability.
2. Multiple sets of experiments are devised to demonstrate the effectiveness of the model, encompassing full supervised settings, few-shot learning setting, multi-task learning setting, and more. The experimental setups are comprehensive, and detailed analyses are conducted for each structural aspect of the model.
3. The writing is clear and facilitates easy comprehension for readers.

**Reasons To Reject:**

1. The connection between bond dimension and adapter rank is unclear, further theoretical explanation is needed.
2. . In the experiments of Adapter Rank in Section 4.3, it seems that it is not a comparison fair with numerical choices of [4, 8, 64]. why not [4, 8, 12] or [4, 16, 64].
3. As per my understanding, stable training in Section 3.3 is a highlight of the paper, and the authors have provided ample theoretical derivation. However, this point has not been further validated in the experimental section.

**Reproducibility:**

4: Could mostly reproduce the results, but there may be some variation because of sample variance or minor variations in their interpretation of the protocol or method.

**Reviewer Confidence:**

3: Pretty sure, but there's a chance I missed something. Although I have a good feel for this area in general, I did not carefully check the paper's details, e.g., the math, experimental design, or novelty.

---

> ### Author Rebuttal · Authors · 2023-08-29
>
> Thanks for your valuable insights and suggestions. In response to your concerns, we have outlined our responses below. Please do not hesitate to share any further inquiries you may have.
>
> R1:
> The bond dimension is calculated within the MPO decomposition based on the matrix structure and cannot be manually adjusted. It is determined by the hyperparameter MPO shape. In contrast, the adapter rank is a tunable hyperparameter designed to control the additional parameter count introduced. These two parameters are not directly interrelated. We will make this clear in the main text of our paper.
>
> R2:
> We appreciate your attention to this issue. The choice of [4, 8, 64] was primarily based on the observation that increasing the adapter rank beyond 8 did not yield significant performance gains in our experiments. The selection aimed to encompass a broad range of values, evaluating the impact across varying levels of adapter rank. Our findings indicated that adapter ranks from 8 to 64 did not substantially enhance performance. We recognize the need to clarify this rationale in our paper and will ensure to provide a more detailed explanation in the main text.
>
> R3:
> Thank you for your insightful observation regarding stable training in Section 3.3. Your feedback is invaluable in highlighting the importance of this aspect. It's worth noting that, without the proposed initialization method, convergence for the MPOBERT_48 model would not be achievable. To address this concern, we will provide additional explanations and thorough validation of this aspect in the main text of our paper.
>
> Typos:
> Thank you for your thorough review. We have diligently reviewed the labeling errors within the table and revised the grammatical errors present in the paragraphs.

---

### Official Review · Reviewer_4FdP · 2023-08-08

**Soundness:** 3

**Excitement:**

3: Ambivalent: It has merits (e.g., it reports state-of-the-art results, the idea is nice), but there are key weaknesses (e.g., it describes incremental work), and it can significantly benefit from another round of revision. However, I won't object to accepting it if my co-reviewers champion it.

**Paper Topic And Main Contributions:**

This paper proposes a parameter-sharing method that can scale pre-trained language models to a deeper depth while maintaining a relative parameter size.

**Questions For The Authors:**

How does parameter-sharing affect pre-training? Does it slow down or speed up the convergence?

How does parameter-sharing bring efficiency improvements?

How does parameter-sharing affect performance?

**Reasons To Accept:**

Large language models are overparameterized models. Sharing parameters across modules make sense.

This paper demonstrates the effectiveness of the model both experimentally and theoretically.

**Reasons To Reject:**

Missing some important experimental results. (loss curve and ppl.). I think that the results on GLUE benchmark are not strong enough to support the claim that deeper models with parameter sharing are better.

I doubt that parameter sharing can really bring efficiency improvement. The authors should provide a more explicit explanation of the "efficiency" of their model. For instance, presenting real-time inference latency, memory consumption, or training time would be more persuasive.

If I am not mistaken, parameter sharing involves reparameterizing the original module without altering the model architecture. However, the authors did not compare models with the same architecture.

**Reproducibility:**

3: Could reproduce the results with some difficulty. The settings of parameters are underspecified or subjectively determined; the training/evaluation data are not widely available.

**Reviewer Confidence:**

3: Pretty sure, but there's a chance I missed something. Although I have a good feel for this area in general, I did not carefully check the paper's details, e.g., the math, experimental design, or novelty.

---

> ### Author Rebuttal · Authors · 2023-08-29
>
> Your insights and suggestions are greatly appreciated. Below, we have provided detailed responses to address your concerns. If you have any additional questions or inquiries, please feel free to share them with us.
>
> R1:
> Thank you for your valuable feedback. In order to highlight the advantages of our approach in terms of increased layer depth with parameter sharing, we have incorporated the loss curves during model training. As illustrated in the table below, we present the validation loss of two models in different steps: (1) MPOBERT_48, representing the deep model with parameter sharing, and (2) MPOBERT_12, representing the shallow model with parameter sharing. By comparing the results of (1) and (2), it becomes apparent that during the initial training stages, the shallow model exhibits lower loss due to better initialization from pre-trained weights across all layers. However, as training progresses, the deep model, owing to its larger model capacity, tends to have a lower loss, showcasing its greater potential during training.  In terms of perplexity, the language modeling perplexity for held-out training data is 6.068 for MPOBERT_12 and 4.855 for MPOBERT_48.
>
> | Model      | 0      | 2000   | 4000   | 6000   | 8000   | 10000  |
> |------------|--------|--------|--------|--------|--------|--------|
> | MPOBERT_48 | 4.368  | 2.160  | 1.844  | 1.698  | 1.583  | 1.580  |
> | MPOBERT_12 | 3.280  | 2.260  | 2.243  | 2.080  | 1.829  | 1.803  |
>
> R2&Q2&Q3:
> We summarize the efficiency of our approach as follows: (1) Real-Time Inference Latency: the MPO-based Transformer's weights can be reconstructed into matrices, which requires a one-time reconstruction before inference. Subsequent inference processes share the same latency as the standard Transformer, without introducing extra computations. (2) Memory Consumption: the adoption of MPOBERT_48 allows for a 48-layer depth with only 16.3G GPU memory, enabling feasible pretraining on a single 24G GPU. (3) Training Time: utilizing 8 V100 GPUs, the training process is completed in just 3.8 days, a substantial reduction compared to the training cost of BERT large. Furthermore, it's noteworthy that MPOBERT_48 outperforms BERT large, demonstrating its superiority with a notable GLUE test score of 81.7, surpassing BERT large's 81.6.
>
> | Model        | #To (M) | Mem (G) | #samples   | #Train time                | GLUE test |
> |--------------|-----|---------|------------|----------------------------|-----------|
> | BERT large   | 330 | -       | 256*1000k  | 16 Cloud TPUs, 4 Days      | 81.6      |
> | MPOBERT_48   | 75  | 16.3    | 4096*10k   | 8 V100 GPUs, 3.8 Days      | 81.7      |
>
> R3:
> It appears that you're referring to the comparison between applying parameter sharing on a MPO-based Transformer layer. Considering the inherent over-parametrization often seen in Transformer models, not implementing parameter sharing results in considerable parameter proliferation as model depth increases. To illustrate, we conducted a comparison involving 12-layer models, designated as "share" (with parameter sharing) and "w/o share" (without parameter sharing). The outcomes reveal that forgoing parameter sharing leads to a substantial 4.9-fold increase in parameters, without yielding proportionate benefits. For instance, in the case of MPOBERT_48, the model features fewer parameters (75M vs. 85M) while exhibiting a lower validation loss (1.583 vs. 1.778).  This outcome underscores that parameter sharing proves more efficient in resource utilization. We appreciate your observation and have discussed this aspect further to provide clarity on the advantages of our approach.
> | Model      | #To (M) | Pre-training steps | Validation Loss |
> |------------|---------|--------------------|-----------------|
> | Share      | 20      | 10k                | 1.803           |
> | w/o Share  | 98      | 10k                | 1.778           |
>
> Q1:
> Please refer to our response ``R3'' to reviewer-mWkV, where we reported the training time and the effects of parameter sharing and initialization. These factors enabled us to train a 48-layer large-scale model in just 3.8 days, which is the shortest training time among all the large-scale models considered.

---

### Official Review · Reviewer_mWkV · 2023-08-12

**Soundness:** 4

**Excitement:**

3: Ambivalent: It has merits (e.g., it reports state-of-the-art results, the idea is nice), but there are key weaknesses (e.g., it describes incremental work), and it can significantly benefit from another round of revision. However, I won't object to accepting it if my co-reviewers champion it.

**Paper Topic And Main Contributions:**

This work introduces a parameter-sharing approach for scaling pre-trained language models (PLMs) to a deeper model depth using the matrix product operator (MPO).



**Questions For The Authors:**

Figure 3 illustrates a continuous improvement in model metrics. Why was the training stopped prematurely?






**Reasons To Accept:**

- The author conducted experiments on multiple datasets, and the results indicate that the proposed method achieves competitive performance compared to the state of the art (SOTA).

- The proposed method appears to have applicability across a wider range of Transformer-based models.

- The author provides theoretical analysis of the proposed method.

**Reasons To Reject:**

- In line 95, the author claims that the proposed method could be applied to various Transformer-based models, yet the experiments are exclusively conducted on the BERT model.
- Some important baseline comparisons are missing, such as RoBERTa-large. Even if the proposed method does not outperform the SOTA baseline, the author should analyze the reasons behind this.
- An analysis of training/inference time is not present.
- The ablation analysis lacks examination of certain crucial hyper-parameters, such as the analysis of length L.





**Reproducibility:**

3: Could reproduce the results with some difficulty. The settings of parameters are underspecified or subjectively determined; the training/evaluation data are not widely available.

**Reviewer Confidence:**

3: Pretty sure, but there's a chance I missed something. Although I have a good feel for this area in general, I did not carefully check the paper's details, e.g., the math, experimental design, or novelty.

---

> ### Author Rebuttal · Authors · 2023-08-29
>
> Thanks for your insightful suggestions and we have listed our response to your concerns as follows. If you also have any other questions, please feel free to let us know. We will continue to try our best to answer for you.
>
> R1:
> We appreciate the reviewers' attention to the scope of our experiments. Our approach focuses on targeted modifications to specific matrices within the Transformer structure, namely the Q,K,V and MLP matrices. These components are foundational to various models based on the Transformer architecture. So, we only chose the most typical BERT for validation, and we will expand our development on more models in the future.
>
> R2:
> While our method has shown improvement over BERT, we acknowledge that surpassing models like RoBERTa-large and T5 have not been achieved. In this study, we used the same training data as BERT (wiki and bookcorpus) and trained for approximately 10k steps over around 3.8 days. The limitations in data availability and training steps could be factors contributing to why our model did not outperform the current best-performing models.
> It's important to note that the core focus of this paper was to contrast against BERT and validate a parameter-efficient structure for building deeper models. Expanding resources and training time is quite time and resource consuming which we intend to explore in our future work, considering the current resource constraints.
>
> R3:
> Thank you for your valuable suggestion. We have incorporated a table presenting the training time and GPU memory usage for the 48-layer MPOBERT in our submission. Utilizing the Deepspeed (DS) training framework with FP16 (fp16) training, our model was trained in just 3.8 days. We will ensure that this table is included in the subsequent sections of the main paper for better visibility and context.
> Copy code
> | Exp.       | BS | Mem (G) | Days  |
> |------------|----|---------|-------|
> | MPOBERT\_48| 4  | 11.6    | 4.9   |
> |            | 8  | 16.3    | 3.8   |
> | w/o DS     | 4  | 11.4    | 5.4   |
> | w/o fp16   | 4  | 18.7    | 12.2  |
> | w/o DS,fp16| 4  | 19.2    | 12.5  |
>
> R4:
> We sincerely appreciate your valuable suggestion. The parameter $L$ represents the length to which the MPO is decomposed.  It's worth noting that during the MPO decomposition process, Liu et al. [1] conducted an extensive study on the effects of varying the length parameter L for MPO decomposition. Their findings indicate that the choice of L has minimal impact on the error between the reconstructed weight matrix obtained through MPO decomposition and the original matrix. In light of your feedback, we will add to this discussion in the main text to offer readers a more comprehensive understanding of the parameter $L$.
>
> Q1:
> We sincerely appreciate your insightful query. In our study, we established a consistent number of pretraining steps to ensure a fair basis for comparison. While Figure 3 showcases the potential for enhanced performance through more pretraining steps, it's crucial to acknowledge that pursuing further steps and utilizing larger datasets demand substantial computational resources. This intriguing avenue for research is undoubtedly a direction we plan to delve into more deeply in the future.
>
> [1] Enabling Lightweight Fine-tuning for Pre-trained Language Model Compression based on Matrix Product Operators

---

### Meta-Review · Area_Chair_FVgq · 2023-09-18

**Recommendation:** 4

**Metareview:**

The paper introduces a novel parameter-sharing method based on the Matrix Product Operator (MPO) for scaling pre-trained language models (PLMs) to deeper model depths. The work emphasizes sharing central tensors of parameter matrices across all layers to achieve efficient deep Transformer architectures. The paper also includes theoretical analysis for why the proposed approach may address observed training instability of deeper PLMs.

Generally, the reviewers liked this approach as it seems broadly applicable to many Transformer-based architectures and there is some consensus that these are often over-parameterized making parameter sharing seem justified. There was some skepticism that parameter sharing can actually lead to improved performance (or similar performance using fewer parameters). The authors provided additional experiments which appeared to satisfy the reviewers.

---

### Decision · Program_Chairs · 2023-10-07

**Decision:**

Accept-Findings

**Comment:**

The paper introduces a novel parameter-sharing method based on the Matrix Product Operator (MPO) for scaling pre-trained language models (PLMs) to deeper model depths. The work emphasizes sharing central tensors of parameter matrices across all layers to achieve efficient deep Transformer architectures. The paper also includes theoretical analysis for why the proposed approach may address observed training instability of deeper PLMs.

Generally, the reviewers liked this approach as it seems broadly applicable to many Transformer-based architectures and there is some consensus that these are often over-parameterized making parameter sharing seem justified. There was some skepticism that parameter sharing can actually lead to improved performance (or similar performance using fewer parameters). The authors provided additional experiments which appeared to satisfy the reviewers.